# Obesity as a Risk Factor for Venous Thromboembolism Recurrence: A Systematic Review

**DOI:** 10.3390/medicina58091290

**Published:** 2022-09-16

**Authors:** Pinelopi Ntinopoulou, Erato Ntinopoulou, Ioanna V. Papathanasiou, Evangelos C. Fradelos, Ourania Kotsiou, Nikolaos Roussas, Dimitrios G. Raptis, Konstantinos I. Gourgoulianis, Foteini Malli

**Affiliations:** 1Department of Cardiology, Laikon General Hospital, 11527 Athens, Greece; 2Department of Pediatrics, Yverdon-les-Bains Hospital, 1400 Yverdon-les-Bains, Switzerland; 3Community Nursing Lab, Faculty of Nursing, University of Thessaly, 41500 Larissa, Greece; 4Faculty of Nursing, School of Health Sciences, University of Thessaly, 41500 Larissa, Greece; 5Department of Vascular Surgery, Faculty of Medicine, University Hospital of Larissa, University of Thessaly, 41500 Larissa, Greece; 6Respiratory Disorders Lab, Faculty of Nursing, University of Thessaly, 41500 Larissa, Greece; 7Respiratory Medicine Department, Faculty of Medicine, University Hospital of Larissa, University of Thessaly, 41500 Larissa, Greece

**Keywords:** obesity, venous thromboembolism, deep venous thrombosis, pulmonary embolism, recurrence, risk factor

## Abstract

*Background and Objectives:* Venous thromboembolism (VTE) encompasses Deep Venous Thrombosis (DVT) and Pulmonary Embolism (PE). The duration of anticoagulant therapy following a VTE event partly relies on the risk of recurrent VTE which depends on the clinical setting where VTE occurred and the VTE risk factors present. Obesity is considered a minor risk factor and studies in the literature have provided conflicting results on whether obesity influences the development of recurrences. The aim of the present study is to assess the effect of obesity on VTE recurrence in patients that suffered from a previous VTE event. *Materials and Methods:* We conducted systematic research for English language studies in Medline, Scopus and ProQuest databases in order to identify publications that assess the risk of VTE recurrence in obesity. Inclusion criteria were: 1. Diagnosis of VTE, 2. Definition of obesity as a body mass index ≥30 kg/m^2^, 3. Report of the risk of obesity on VTE recurrence, 4. Adult human population. We did not include case reports, review studies or studies that assessed other forms of thrombosis and/or used other definitions of obesity. We used the Newcastle-Ottawa scale to address the quality of the studies. *Results:* Twenty studies were included in the analysis, of which 11 where prospective cohort studies, 6 were retrospective cohort studies, 1 was a cross-sectional study, and 2 were post-hoc analysis of randomized clinical trials. Obesity was significantly associated with recurrences in 9 studies and in 3 of them the association was significant only in females. *Conclusions:* There is heterogeneity between the studies both in their design and results, therefore the effect of obesity on VTE recurrence cannot be adequately estimated. Future randomized clinical studies with appropriately selected population are needed in order to streamline the effect of obesity on VTE recurrence.

## 1. Introduction

Venous thromboembolism (VTE) encompasses two interrelated medical conditions that include deep vein thrombosis (DVT) and pulmonary embolism (PE) [1]. VTE results from an interaction of the patients’ risk factors and the clinical setting where the event occurs. Virchow’s Triad distills the large number of VTE risk factors into three basic pathophysiological mechanisms that may result in thrombus formation: venous stasis, endothelial injury, and hypercoagulability [2]. The clinical presentation of VTE ranges from absence of symptoms to hemodynamic instability (massive PE) that may result in death. Mortality is mainly attributed to patients suffering from PE and may be as high as 30% in undiagnosed, and thus, untreated cases [3]. VTE management in the acute phase (i.e., first hours to days after diagnosis) varies according to the severity of the event from oral anticoagulants to reperfusion therapies [4]. Current guidelines recommend at least 3 months of anticoagulation for all patients with VTE, irrespectively of the nature and clinical setting of the episode [4,5]. The decision to recommend extended treatment (beyond 3 months) depends on the risk of recurrence and bleeding [6]. Patients with major permanent risk factors (e.g., patients with antiphospholipid antibody syndrome, patients with active cancer) have increased recurrence risk and extended treatment is generally recommended. On the other hand, patients with major transient risk factors (e.g., surgery, hip fracture) typically discontinue anticoagulation beyond the long-term phase of therapy. In all other cases, where intermediate or minor risk factors are present, the recommendation of extended treatment varies, since the individual risk of recurrence cannot be effectively estimated.

Obesity is defined as the presence of excessive fat accumulation that presents a well-recognized health risk [7]. A body mass index (BMI) ≥30 kg/m^2^ is commonly used as cut-off for obesity definition. Obesity is considered as a minor risk factor that exerts more than a 2-fold increased risk of a first episode of VTE [8]. However, the impact of obesity on VTE recurrence remains ambiguous since the published studies present controversial results and have methodological disparities. The estimation of the risk associated with obesity is important since it could help in the assessment of the individual risk of recurrence and thus the recommendation on extended anticoagulation. The objective of the present study was to evaluate the risk of VTE recurrence in obese subjects in comparison to non-obese VTE population. To this end, we conducted a systematic review of the literature that investigates the incidence and the risk of VTE recurrence (DVT and/or PE) in obese patients.

## 2. Materials and Methods

The present review was performed according to the Preferred Reporting Items for Systematic Reviews and Meta-analysis (PRISMA) guidelines [9] (Appendix A). The study protocol is registered with the PROSPERO international prospective register of systematic reviews (protocol number: CRD42022349562).

### 2.1. Search Strategy

For the present systematic review, we performed a literature search in MEDLINE via PubMed, Scopus and ProQuest databases for relevant studies. The search algorithm was as follows: (obes* OR BMI) AND recur* AND ((deep vein thrombosis) OR (pulmonary embolism) OR (thromboembol*)). The keywords should be included in the title and/or summary and/or keywords of the study for MEDLINE and Scopus and anywhere in the text for ProQuest database. The search was performed at the 11 January 2022 for MEDLINE and Scopus databases and at the 20 July 2022 for the ProQuest database. The results were screened and duplicate studies were removed. Two reviewers (PN and FM) independently reviewed the titles and abstracts of all candidate studies that were identified by the search. Studies that were irrelevant to the research question were excluded. The articles extracted by the two reviewers were subjected to full-text review and were assessed for the eligibility criteria. For relevant studies, the full texts were obtained and evaluated for pertinent information. We also checked the full text articles for references of interest (i.e., snowballing). The agreement about article selection among the two reviewers was excellent (k ¼ 0.83). When conflicts in study selection arose, they were resolved by consensus, and, if failure to reach an agreement occurred a third author (KIG) was consulted.

### 2.2. Selection Criteria

Inclusion criteria for the studies were: 1. Objectively confirmed diagnosis of DVT and/or PE according to internationally published guidelines [4,5], 2. International Classification of Diseases (ICD) coded diagnosis of DVT and/or PE, 3. Definition of obesity according to BMI, i.e., BMI ≥ 30 kg/m^2^ [7], 4. Report of the risk of obesity on recurrence of VTE (DVT and/or PE), 5. Adult human population, 6. English language. We excluded studies that assessed superficial vein thrombosis of upper and/or lower limbs, splanchnic vein thrombosis, arterial thrombosis, studies that defined obesity based on waist circumference, in vitro studies, animal studies, review studies and case reports. We used AXIS for the quality assessment of the cross-sectional studies [10], the Newcastle-Ottawa Quality Assessment Scale (NOS) for the cohort and case-control studies [11] and the Cochrane collaboration risk of bias tool (CCRBT) for the randomized controlled trials [12]. For the NOS three quality parameters were analyzed: study selection, comparability of the population and determination of whether the exposure or outcome includes risk of bias. The studies were considered as high quality if they scored ≥7, moderate if they scored 5/6, and low if they scored <5.

### 2.3. Outcome and Data Extraction

The primary outcome question was the effect of obesity on the risk of VTE recurrence in patients that suffered from a previous VTE event. Thus, the main outcome assessed was the risk of VTE recurrence (i.e., objectively confirmed PE and/or DVT or ICD coded diagnosis of PE and/or DVT in a patient with a history of previous VTE) in VTE patients with a BMI ≥ 30 kg/m^2^ [7]. Eligible studies were assessed independently by two authors (PN, FM). Study design (year of publication, methods used, geographical area), study population and demographics (age distribution, gender distribution, sample size, rate of obese subjects), and outcome measure (DVT, PE, use of anticoagulants) were extracted. The definition of obesity was a BMI ≥ 30 kg/m^2^. Use of anticoagulants was defined as use of unfractioned heparin, low molecular weight heparin, fondaparinux, Vitamin K antagonists or direct oral anticoagulants. We defined provoked and unprovoked events according to internationally accepted criteria [13].

Ethical approval or patient consent was not required since the study was based on the data of previously published studies.

## 3. Results

### 3.1. Study Selection

The search strategy revealed a total of 2937 studies (356 indexed in MedLine, 763 indexed in Scopus, 1818 indexed in ProQuest). After removal of duplicates and screening of titles, abstracts and keywords, 115 studies underwent full-text review. Of these 115 studies, along with the results of snowballing, a total of 29 studies met the inclusion criteria. After the full text review, 20 studies were included in the analysis. The PRISMA flow diagram of the selection process is presented in Figure 1.

The studies’ characteristics are presented in Table 1. The 20 studies are presented in chronological order [14,15,16,17,18,19,20,21,22,23,24,25,26,27,28,29,30,31,32,33]. As far as study design is concerned, 11 studies where prospective cohort studies [14,16,17,18,20,21,22,26,27,31,33], 6 were retrospective cohort studies [19,23,25,28,29,32], 1 was a cross-sectional study [15] and 2 were post-hoc analysis of randomized clinical trials [24,30]. Seven of the studies were performed in USA [23,25,26,28,29,32,33], 2 in Spain [14,22], 1 in Italy [18], 1 in France [20], 1 in Switzerland [27], 1 in Austria [17], 1 in Iran [19], and six were multicenter studies [16,21,24,30,31,33]. Fourteen of the studies included patients that had a mixed VTE population (PE and/or DVT) [14,15,16,17,18,20,21,22,24,26,27,32,33], 4 studies included patients with isolated DVT [19,25,29,31] and 2 studies included only PE patients without DVT [28,30]. In 4 of the studies, VTE was considered unprovoked [17,20,21,22]. In 7 studies, patients were included after the discontinuation of anticoagulants [14,16,17,20,21,22,26], in 4 studies, the population was under anticoagulant therapy [24,31,32,33] while in 1 study patients were studied both during and after anticoagulant discontinuation [30]. The results of the studies are summarized in Table 2. The quality assessment and risk of bias of the included studies is presented in Appendix A according to the study design.

### 3.2. Obesity as Risk Factor for VTE Recurrence

The studies are analyzed and grouped according to their design.

Eleven of the studies were prospective cohort studies. In 3 of the studies, there was a positive significant correlation of obesity with VTE recurrence [17,18,22]. Eichinger et al. [17] and Moreno et al. [22] studied subjects with a BMI ≥ 30 kg/m^2^ and unprovoked DVT and/or PE whose diagnosis was confirmed objectively. In both studies, patients had discontinued anticoagulants during follow-up. Multivariate analysis was performed, after adjustment for factors such as gender, age and hereditary thrombophilia, that demonstrated that obesity is positively associated with VTE recurrence. One has to take into account that the follow-up time was not adequately described in one study [17]. Additionally, Moreno et al. [22] did not perform adjustments for every hereditary thrombophilia factor, while the follow-up time was less than two years, which may be considered limited for the assessment of recurrences. Di Nisio et al. [18] studied VTE patients which were further stratified according to their muscle strength (normal or poor). After adjusting for gender, age, cardiovascular disease, hip fracture, previous hospitalization (within 1 year) and fitness status, researchers observed a significant positive correlation of obesity with VTE recurrence in patients with both normal and poor muscle strength. In the remaining 4 prospective cohort studies, obesity was not associated with VTE recurrence [14,26,27,33]. Garcia-Fuster et al. [14] and Vučković et al. [26] did not find a significant correlation of obesity with recurrence, in VTE (objectively confirmed DVT and/or PE) obese patients following their anticoagulant withdrawal and after adjustment for gender, age and site of VTE. Potential criticisms include a small sample size (<100 patients) [14] and poor recording of BMI (i.e., BMI was assessed only in the inclusion to the study, weight and height were patient reported and not objectively confirmed) [26]. Weitz et al. [33] studied patients with DVT and/or PE from 28 countries that were receiving anticoagulant therapy and did not observe a significant correlation of obesity with VTE recurrence. The study included a large sample size and VTE diagnosis was based on ICD code. Additionally, Mueller et al. [27], analyzed data from VTE patients with a BMI ≥ 30 kg/m^2^ after adjustment for gender, age, type of VTE, personal and family history, and did not find a significant association of obesity with recurrence.

Six studies had retrospective cohort design. The same research team studied a large sample of DVT [29] and PE patients [28] and demonstrated a significant positive correlation of obesity with VTE recurrence. There are no data about the use of anticoagulants during the follow-up period [28,29]. Cardinal et al. [32], studied a smaller sample of patients that were receiving anticoagulants and did not observe a significant association of obesity with recurrence. The retrospective studies mentioned earlier used ICD coding for the inclusion in the study which may render some uncertainty about the diagnosis of VTE. The fact that the use of anticoagulants during follow-up was not recorded, poses a limitation in the results of the studies [28,29].

In the only cross-sectional study, there was not a significant association of obesity with VTE recurrence after multivariate analysis and adjustment for age, gender, cancer, surgery, hormone therapy, inflammatory diseases, immobilization and hereditary thrombophilia [15].

Di Nisio et al. [24] and Beenen et al. [30] performed a post hoc analysis of large randomized controlled trials of patients with objectively confirmed VTE [21] and PE [30] that were studied while receiving anticoagulants [24,30] and after their withdrawal [30]. Both of the studies failed to demonstrate a significant association of obesity with VTE recurrence. Major caveats that should be mentioned include the fact that the number of recurrences where small, BMI was objectively recorded only during the entry of the study and multivariate analysis was performed after adjustment only for age, gender, cancer and glomerular filtration rate.

Two of the studies included patients with isolated DVT without PE. Asim et al. [25] performed a retrospective cohort study in patients with objectively confirmed DVT and observed that obesity may represent a risk factor for recurrence. In multivariate analysis they adjusted for age, gender, cancer, diabetes mellitus, coagulopathies, history of PE and bilateral DVT, but did not provide information about the use of anticoagulants during the study period. Farzamnia et al. [19] performed a retrospective cohort study in Iran in patients with objectively confirmed DVT and included a relatively small number of obese subjects. Obesity was not associated with DVT recurrence which may be attributed to the small proportion of obese patients included in the study (1.8%).

#### 3.2.1. Analysis according to Gender

In 3 of the prospective cohort studies, obese patients were stratified according to gender [16,20,21]. Two of the studies were multicenter [16,21] and the third was performed in France [20]. VTE diagnosis was objectively confirmed and patients were not receiving anticoagulant treatment. Studies showed a positive significant association of obesity with VTE recurrence in females but not in males. It should be mentioned that in the study of Rodger et al. [16] the number of obese subjects included is small, a fact that may limit the strength of the study results.

#### 3.2.2. Analysis according to Cancer History

In two of the studies, researchers evaluated obesity as a risk factor for recurrence in patients with active cancer. Huang et al. [23] performed a retrospective cohort study in USA of patients with VTE diagnosis according to ICD coding, with and without active cancer. Giorgi-Pierfranceschi et al. [31] studied prospectively a cohort of patients with objectively confirmed DVT and a BMI ≥ 40 kg/m^2^. Patients were receiving anticoagulant therapy and were stratified according to the presence of active cancer. In both studies, obesity was not associated with VTE recurrence. Arguments that can be made include the differences in data recording methodology between centers, the recording of BMI at study inclusion and the use of DOACS in patients with a BMI ≥ 40 kg/m^2^ in which the effectiveness and safety is questioned [31,34].

#### 3.2.3. Analysis according to the Use of Anticoagulants

Seven prospective cohort studies included subjects with objectively confirmed VTE that were not receiving anticoagulants during follow-up. Two of them demonstrated that obesity is a risk factor for recurrences in both males and females [17,22] while in others the relationship prevailed in the female population [16,20,21]. In two of the studies researchers failed to demonstrate a relationship of obesity with VTE recurrence [14,26]. In 4 of the studies that patients were receiving anticoagulants during follow-up there was not a significant relationship of obesity with recurrences [24,31,32,33].

#### 3.2.4. Analysis according to the Presence of other Provoking Factors

All the studies that included exclusively subjects with unprovoked VTE demonstrated a significant association of obesity with recurrences [17,20,21,22]. In two of the studies [17,18] the association was significant only in female patients. Τhis result raises questions since in most of the studies the clinical setting in which VTE event occurred is not mentioned.

## 4. Discussion

In the present systematic review, we have analyzed the results of 20 carefully selected studies that address the role of obesity on VTE recurrence. The findings of the studies are contradictory with 9 of them demonstrating a significant association of obesity with recurrence of VTE. The relationship was evident in only the female population in 3 studies. In general, the quality of evidence was low suggesting the need for further well-designed studies in order to elucidate the research question.

Defining the impact of obesity on VTE recurrence will become increasingly critical. Obesity presents a major public health issue and its’ prevalence in the Western world has increased in recent years. In USA, approximately 1/3 of the adult population is obese [35]. According to the World Health Organization (WHO), in 2016, 13% of the adult population was obese [36]; this corresponds to 3 times increase since 1975. If the current trend continues, by 2025 WHO estimates that 1 billion adult population worldwide will be obese. On the other hand, VTE prevalence has risen [37]. Obesity has been consistently associated with a higher prevalence of PE and/or DVT and has been recognized as a risk factor for VTE [38]. Collectively, these findings highlight the necessity to elucidate the potential impact of obesity in deciding whether to extend anticoagulant treatment in VTE patients since it will become an increasingly important and common issue.

The results of the studies analyzed in the present systematic review were conflicting. Obesity is considered a risk factor for a first VTE episode, however its’ role in developing a second event is not well understood. Although this seems like a paradox, studies have shown that the risk profile of a recurrent VTE is not the same as the first event [39]. In a meta-analysis, obesity has been associated with a 2.33-fold increased risk of developing VTE [40]. Interestingly, obesity has been associated with the occurrence of a first distal DVT event [41]. However, obesity is not associated with VTE recurrence in a population of distal DVT patients [42] suggesting that there is no equality of results in this population too. Here, we observed a significant association of obesity with VTE recurrence in 9 out of the 20 studies included in the analysis. Odds Ratio ranged from 1.53–9.69 [18,29] and Hazard Ratio for the female population was 2.3–2.8 [17,20]. Significant heterogeneity was observed within studies that makes direct comparison of their results difficult.

The factors contributing to VTE development are multiple and are summarized in the 3 factors of Virchows’ triad: endothelial injury/trauma, hypercoagulability and venous stasis [2]. The mechanism(s) underlying the relationship between excess body weight and VTE are poorly defined. Mechanistic reasons associated with obesity may result in reduced venous return [43]. Increased BMI is associated with elevated prothrombotic molecules such as Factor VII, fibrinogen and tissue factor, possibly through changes in systemic inflammation [44]. Additionally, plasminogen activator inhibitor-1 (PAI-1) is consistently elevated in obesity, possibly through increased PAI-1 expression in visceral tissue [45]. In this context, obesity may result in impaired fibrinolysis. The presence of possible confounders associated with obesity, like diabetes, reduced physical activity or diet, may favor the development of VTE in the obese population [43]. One has to take into account that the contemporary definition of obesity according to WHO depends on BMI [7] which corresponds well to body fat estimation but cannot effectively reflect fat distribution and/or abdominal adiposity [46]. Other indicators such as waist circumference may be better predictors of cardiometabolic risk. Researchers have suggested that waist circumference may represent the preferred measure of obesity when considering VTE risk [47], however it lacks reproducibility [48].

Some limitations of the studies included in the present systematic review should be taken into account when analyzing the data. Potential criticisms include that in some studies, BMI was assessed only at study inclusion and potential weight change during follow-up was not addressed. Additionally, none of the studies investigated adherence to anticoagulants that could potentially affect the rate of VTE recurrence. The clinical implication of the findings of the present systematic review cannot be adequately evaluated due to the design of the studies included. In order to estimate the clinical application of obesity in deciding whether to extend anticoagulant treatment in VTE, further randomized clinical trials with appropriately-matched obese and non-obese population are warranted. The short follow-up time in most studies poses another limitation to their results. The relevance of other risk factors for recurrence was not appropriately examined in the studies included, since multivariate analysis was not performed in all of them or was performed for limited factors (i.e., gender, age, hereditary thrombophilia, type of VTE, family history, cancer, surgery, hormone therapy, inflammatory diseases, immobilization) which differed between studies; therefore, we cannot sufficiently measure the independent risk that obesity poses to developing a second event.

## 5. Conclusions

The available literature concerning obesity as a risk factor of recurrence is limited and heterogenous while the quality of evidence is low. The present systematic review provides conflicting results as to whether obesity increases the risk of deve loping a subsequent VTE event. The role of obesity on VTE recurrence needs to be evaluated more rigorously in order to adequately and precisely address its’ impact on the decision to extend anticoagulation.

## Figures and Tables

**Figure 1 medicina-58-01290-f001:**
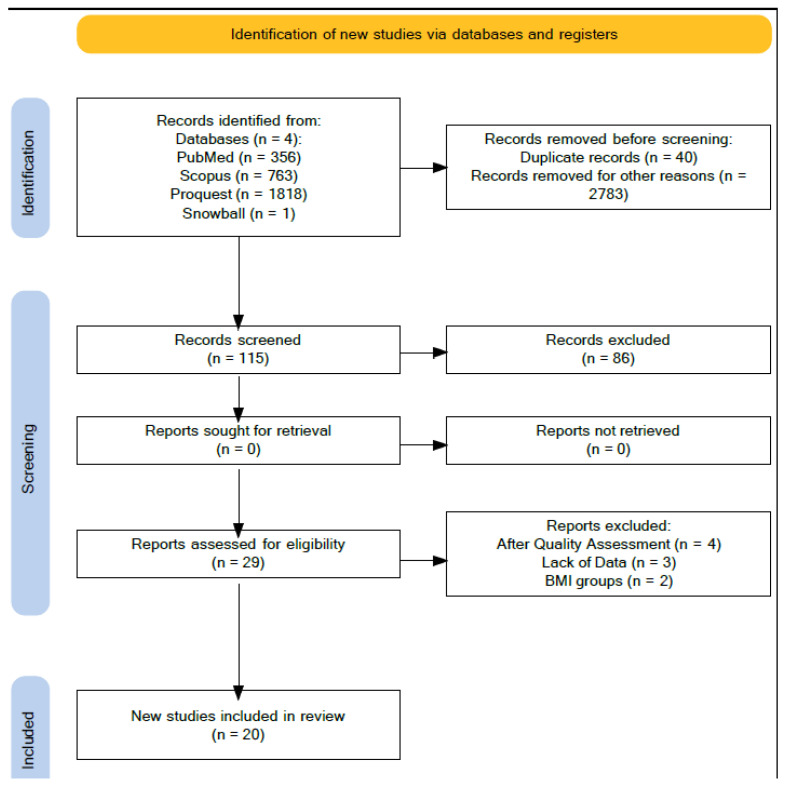
PRISMA flow diagram of the selection process.

**Table 1 medicina-58-01290-t001:** The studies’ characteristics. Objectively confirmed VTE diagnosis (PE or DVT) was based on clinical and imaging data. “Anticoagulants during follow-up” column refers to whether the subjects enrolled in the study were receiving anticoagulants when the recurrence occurred after the first VTE event. Abbreviations: BMI (kg/m^2^); VTE: Venous thromboembolic disease; DVT: deep venous thrombosis; PE, pulmonary embolism; m, mean; md, median, y, years; mo, months; n, absolute number; ICD, International Classification of Diseases; NA: Not available or not applicable., * Data are presented as mean ± SE, Data are presented as mean (range).

Reference	First Author, Year Published	Type of Study	Country (Study Duration)	Follow-up Time (Months) (m or md)	Sample Size (n)	Age (Years) [Mean ± SD or Median (Range)]	Male/Females (n (%))	Obese Subjects Included (n (%))	Primary Event (Diagnostic Criteria)	Provoked/Unprovoked Event	Anticoagulants During Follow-up
[14]	Garcia-Fuster M.J., 2005	Prospective cohort	Spain (1989–2004)	117 (md)	98	32.29 ± 9.2	50/48 (51.02%/48.98%)	18 (18.36%)	VTE (objectively confirmed)	NA	No
[15]	Linnemann B., 2008	Cross-sectional	USA (2000–2006)	NA	1006	43.5 ± 17	424/582 (42.15%/57.85%)	226 (22.7%)	VTE (objectively confirmed)	NA	NA
[16]	Rodger M.A., 2008	Prospective cohort	multicenter (4 counries) (2001–2006)	18 (m)	314	53 (18–95)	0/314 (0%/100%)	28 (8.9%)	VTE (objectively confirmed)	NA	No
[17]	Eichinger S., 2008	Prospective cohort	Austria (1992–2006)	46 (m)	1107	49 ± 16	518/589 (46.80%/53.20%)	271 (24%)	VTE (objectively confirmed)	100% unprovoked	No
[18]	Di Nisio M., 2011	Prospective cohort	Italy (1998–2000)	72	1045	74.6 ± 0.8 (VTE), 74 ± 0.2 (no VTE) *	462/583 (44.21%/55.79%)	265 (25.35%)	VTE (objectively confirmed)	NA	NA
[19]	Farzamnia H., 2011	Retrospective cohort	Iran (2000–2011)	NA	385	48.3 ± 19.16	228/157 (59.22%/40.78%)	7 (1.81%)	DVT (objectively confirmed)	NA	NA
[20]	Olié V., 2012	Prospective cohort	France (2003–2009)	28 (m)	583	45.8 ± 19.7 (females), 54.4 ± 14.8 (males)	234/349 (40.14%/59.86%)	114 (19.55%)	VTE (objectively confirmed)	100% unprovoked	No
[21]	Rodger M.A., 2016	Prospective cohort	Multicenter (4 countries) (2001–2006)	60 (m)	663	53.2 (18–95)	341/322 (51.43%/48.57%)	248 (37.5%)	VTE (objectively confirmed)	100% unprovoked	No
[22]	Franco Moreno A.I., 2016	Prospective cohort	Spain (2004–2013)	21.3 (md)	398	61 (md)	217/181 (54.52%/45.48%)	111 (27.8%)	VTE (objectively confirmed)	100% unprovoked	No
[23]	Huang W., 2016	Retrospective cohort	USA (1999–2009)	30 (m), 23.4 (md)	2989	64.3 ± 18. 67	1319/1670 (44.13%/55.87%)	826 (27.63%)	VTE (ICD coded)	NA	NA
[24]	Di Nisio M., 2016	Post-hoc analysis of a RCT	Multicenter (30 countries) (2007–2009)	NA	8230	56 (18–97)(BMI < 25), 60 (18–97) (25 ≤ BMI < 30), 60 (18–97) (30 ≤ BMI < 35), 55 (20–92) (BMI ≥ 35)	4489/3741 (54.54%/45.46%)	2491 (30.26%)	VTE (objectively confirmed)	64% unprovoked	Yes
[25]	Asim M., 2017	Retrospective cohort	USA (2008–2012)	12	662	50 ± 17	325/337 (49.09%/50.91%)	257 (47%)	DVT (objectively confirmed)	NA	NA
[26]	Vučković B.A., 2017	Prospective cohort	USA (1999–2004)	67.2 (md)	3889	49 (18–70)	1750/2139 (45.00%/55.00%)	814 (20.93%)	VTE (objectively confirmed)	2688/1155	No
[27]	Mueller C., 2017	Prospective cohort	Switzerland (2009–2013)	28 (m)	986	75 (69–81)	526/460 (53.35%/46.65%)	242 (24.54%)	VTE (objectively confirmed)	NA	NA
[28]	Stewart L.K., 2020	Retrospective cohort	USA (2004–2017)	66 (md)	72,936	58 (m)	32,821/40,115 (45.00%/55.00%)	16,046 (22%)	VTE (NA)	NA	ΝA
[29]	Stewart L.K., 2020	Retrospective cohort	USA (2004–2017)	24	151,054	58 (m)	66,464/84,590 (44.00%/56.00%)	28,700 (19%)	PE (ICD coded)	NA	ΝA
[30]	Beemen L.F.M., 2020	Post-hoc analysis of a RCT	Multicenter (37 countries) (2010–2012)	12	1911	56.9 ± 16.6	NA	672 (35.16%)	DVT (ICD coded)	NA	Yes and No
[31]	Giorgi-Pierfranceschi M., 2020	Prospective cohort	Multicenter (27 countries) (2001–2018)	NA	16,490	64 ± 12 (BMI ≥ 40, patients with cancer), 67 ± 14 (BMI = 18.5–24.9, patients with cancer), 59 ± 16 (BMI ≥ 40, patient without cancer), 61 ± 22 (BMI = 18.5–24.9, patients without cancer)	7678/8812 (46.56%/53.44%)	1642 (9.95%)	PE (objectively confirmed)	NA	Yes
[32]	Cardinal R.M., 2021	Retrospective cohort	USA (2012–2017)	12	1059	NA	555/504 (52.40%/47.60%)	552 (52.12%)	DVT (objectively confirmed)	NA	Yes
[33]	Weitz J.I., 2021	Prospective cohort	Multicenter (28 countries) (2014–2017)	24	9479	61.9 (BMI < 18.5), 59.6 (BMI = 18.5–24.9), 61.2 (BMI = 25–29.9), 58.9 (BMI ≥ 30) (median)	4772/4707 (50.34%/49.66%)	3073 (32.41%)	VTE (ICD coded)	NA	Yes

**Table 2 medicina-58-01290-t002:** The studies’ results. Abbreviations: BMI (kg/m^2^); VTE: Venous thromboembolic disease; DVT: deep venous thrombosis; PE, pulmonary embolism; m, mean; md, median, y, years; mo, months; n, absolute number; NA: Not available or not applicable., Data are presented as mean ± SE, Data are presented as mean (range).

Reference	Title	Type of Study	Sample Size (n)	Obese Subjects (n(%))	OR/HR/RR (95%, CI) for Recurrence for Obese Subjects	Association of Obesity with VTE Recurrence
García-Fuster et al. [14]	Long-Term Prospective Study of Recurrent Venous Thromboembolism in Patients Younger than 50 Years	Prospective cohort	98	18 (18.36%)	RR = 1.92 (0.83–4.43)	Not significant
Linnemann et al. [15]	Impact of sex and traditional cardiovascular risk factors on the risk of recurrent venous thromboembolism: results from the German MAISTHRO Registry	Cross-sectional	1006	226 (22.7%)	RR = 1.1 (0.83–1.35), *p* = 0.664	Not significant
Rodger et al. [16]	Identifying unprovoked thromboembolism patients at low risk for recurrence who can discontinue anticoagulant therapy	Prospective cohort	646	28 (8.9%)	RR = 2.33 (1.14–4.74), *p* = 0.02	Significant association (females only)
Eichinger et al. [17]	Overweight, obesity, and the risk of recurrent venous thromboembolism	Prospective cohort	1107	271 (24%)	HR = 1.6(1.0–2.4), *p* = 0.02	Significant association
Di Nisio et al. [18]	Obesity, poor muscle strength, and venous thromboembolism in older persons: the InCHIANTI study	Prospective cohort	1045	265 (25.3%)	OR = 9.69 (3.13–30.01) for obesity, OR = 14.57 (5.16–41.15) for obesity with reduced mauscle strength	Significant association
Farzamnia et al. [19]	The Predictive Factors of Recurrent Deep Vein Thrombosis	Retrospective cohort	385	7 (1.81%)	OR = 0.013, *p* = 0.908	Not significant
Olié et al. [20]	Sex specific risk factors for recurrent venous thromboembolism.	Prospective cohort	583	114 (19.55%)	HR = 2.8 (1.3–6) (Female population)	Significant association (females only)
Rodger et al. [21]	Long-term risk of venous thrombosis after stopping anticoagulants for a first unprovoked event: A multi-national cohort	Prospective cohort	663	248 (37.5%)	OR = 2.3 (1.1–5.1) (female populaton)	Significant association (females only)
Franco Moreno et al. [22]	A risk score for prediction of recurrence in patients with unprovoked venous thromboembolism (DAMOVES)	Prospective cohort	398	111 (27.8%)	HR = 3.92 (1.75–8.75), *p* = 0.0001	Significant association
Huang et al. [23]	Occurrence and predictors of recurrence after a first episode of acute venous thromboembolism: population-based Worcester Venous Thromboembolism Study	Retorpsective cohort	2989	826 (27.63%)	HR = 0.81 (0.7–1.21) at 3 years, HR = 0.70 (0.43–1.13). At 3 months, HR = 0.46 (0.15–1.44) patients with active cancer, HR = 0.79 (0.46–1.36) patients without active cancer	Not significant
Di Nisio et al. [24]	Treatment of venous thromboembolism with rivaroxaban in relation to body weight. A sub-analysis of the EINSTEIN DVT/PE studies	Post-hoc of a RCT	8230	2491 (30.26%)	HR = 0.70 (0.31–1.57) f for patients with BMI = 30–34.9: HR = 1.45 (0.62–3.39) for patients with BMI ≥ 35:	Not significant
Asim et al. [25]	Recurrent Deep Vein Thrombosis After the First Venous Thromboembolism Event: A Single-Institution Experience	Retrospective cohort	662	257 (47%)	OR = 2.2 (1.37–3.53), *p* = 0.001	Significant association
Vučković et al. [26]	Recurrent venous thrombosis related to overweight and obesity: results from the MEGA follow-up study	Prospective cohort	3889	814 (20.93%)	HR = 1.05 (0.85–1.30)	Not significant
Mueller et al. [27]	Obesity is not associated with recurrent venous thromboembolism in elderly patients: Results from the prospective SWITCO65+ cohort study	Prospective cohort	986	242 (24.51%)	HR = 1.10 (0.7–1.74)	Not significant
Stewart et al. [28]	Metabolic Syndrome Increases Risk of Venous Thromboembolism Recurrence after Acute Pulmonary Embolism	Retrospective cohort	72,936	16,046 (22%)	HR = 2.08 (2.00–2.17)	Significant association
Stewart et al. [29]	Metabolic syndrome increases risk of venous thromboembolism recurrence after acute deep vein thrombosis	Retrospective cohort	151,054	28,700 (19%)	OR = 1.53 (1.48–1.59)	Significant association
Beemen et al. [30]	Prognostic characteristics and body mass index in patients with pulmonary embolism: does size matter?	Post-hoc of a RCT	1911	672 (35.16%)	OR = 1.82 (0.78–4.25) for BMI = 30–34.9: OR = 0.71 (0.78–3.34), for BMI =35–39.9 OR = 1.41 (0.78–2.53) for BMI ≥ 40	Not significant
Giorgi-Pierfranceschi et al. [31]	Morbid Obesity and Mortality in Patients With VTE: Findings From Real-Life Clinical Practice	Prospective cohort	16,490	1642 (9.95%)	HR = 1.01 (0.64–1.58) for BMI ≥ 40 without cancer: HR = 1.03 (0.52–2.01) for BMI ≥ 40 with cancer	Not significant
Cardinal et al. [32]	Safety and efficacy of direct oral anticoagulants across body mass index groups in patients with venous thromboembolism: a retrospective cohort design	Retrospective cohort	1059	552 (52.19%)	OR = 0.98 (0.49–1.65) for BMI = 30–39.9 OR = 1.52 (0.74–3.15) for BMI ≥ 40	Not significant
Weitz et al. [33]	Influence of body mass index on clinical outcomes in venous thromboembolism: Insights from GARFIELD-VTE	Prospective cohort	9479	3073 (32.41%)	HR = 1.07(0.85–1.340), *p* = 0.5521	Not significant

## Data Availability

Not applicable.

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
