# Peer review of "Obesity as a Risk Factor for Venous Thromboembolism Recurrence: A Systematic Review"

_medicina, 2022, doi:10.3390/medicina58091290_

Round 1

Reviewer 1 Report

Thankyou very much for opportunity to review this manuscript. I have a few comments for improvement of this manuscript. 

1. Usually, a systematic review uses at least 3 or 4 databases for the literature search. In this manuscript, the Authors use only 2 databases (Medline and Scopus). I highly recommend the Authors to expand the literature search using other databases (Embase, Web of Science, Proquest, etc) to minimize the chance of articles not included in the systematic review.

2. In recent years, it has become increasingly important for a systematic review to be registered in PROSPERO (International prospective register of systematic reviews) to avoid redundant or too similar systematic review. I recommend the Authors to register the systematic review retrospectively to PROSPERO and provide ID number of the registration in the revision.

3. The information on search strategy used should be improved. Kindly see the examples from https://www.mdpi.com/1648-9144/58/8/1058/htm and https://www.mdpi.com/1648-9144/58/8/1069/htm from the same journal the Authors is submitting

4. I highly recommend the Authors use reporting guideline from PRISMA checklist for the manuscript. Revise based on PRISMA checklist and provide the checklist as a supplement.

5. In Table 1, 2, and 3, I recommend instead of listing the studies using numbers only such as [11], [12], etc the Authors should also include the first author names. For examples, Inazaki et al. (2016) [11] and Einstein et al. (2020) [12}

6. Kindly review the title of the manuscript as it seems there is an extra unnecessary word of “title”

7. In the method section, the Authors wrote that “…….CASP Checklist for the cohort and case-control studies”. However, the abstract and the result section used Newcastle Ottawa scale and there is no mention of CASP checklist or evaluation of studies using CASP checklist. Kindly confirm again about the checklist used in the manuscript and revise as needed.

Overall, I believe the manuscript requires major revisions before publication. The manuscript does have solid foundation but improvements are needed in order for the manuscript to achieve its maximal potential. I have confidence that the Authors can conduct the revisions listed by me and other reviewer(s). I sincerely thank the Authors for their work in the manuscript. 

Reviewer 2 Report

INTRODUCTION

Line 62 – do the authors mean acute phase of therapy?

METHODS

Section 2.1 Suggest supplementary material with clearly defined search strategies

Section 2.3 – pls be clearer re the outcome measures. For instance, what does the use of anticoagulants mean? How were recurrent events diagnosed? Were recurrences while on or off therapy?

How was assessment of risk of bias performed?

Figure 1 – “results sought for retrieval” and “reports not retrieved” – what does NA means? How did 91 studies become 29 studies?

What does quality assessment mean? Is this subjective?

RESULTS

Line 161 – “six were international studies” – do you mean multi-centre? Also mentioned in line 215

Table 1 – last column – what does anticoagulants during follow up mean?

I find it hard to draw anything conclusive from studies #11, 13 and 16 given how few obese patients there are.

Pls read through the paper and correct the grammatical and spelling mistakes.

-       Eg line 168, no need for comma after researchers.

-       Line 183: Two of them, that included…

-       Line 240: demonstrated

-       Line 275: prothrombotic

What does it mean when the authors describe that the patients are not under anticoagulant treatment eg lines 216-217 (mentioned several times throughout the manuscript) – do you mean anticoagulation has been ceased following completion of acute phase therapy?

Line 292 – rather than PE, do the authors mean VTE?

Reviewer 3 Report

I read with great interest this present systematic review about obesity as a risk factor for recurrent VTE.

In general, the manuscript addresses a clinically relevant issue in this increasing population of patients worldwide. The study design is very clear and ambitious.

Moreover, several critical revisions should be made to improve the paper before publication.

Minor revision

-       Could the author explain why splanchnic veins thrombosis were excluded from this study about VTE. Did portal or mesenteric veins were also excluded?

-       Table 1. There is several typographic errors. This table should be improved in particular with % of men and women (not only n=).

-       In only few studies, non-provoked VTE was described. Did the authors review each paper looking for the presence or the absence of major risk factor of VTE to redefine precisely provoked or non-provoked VTE in the included studies (Refer to Kearon C, JTH 2016)? If not, it should be, because unprovoked VTE in obese patients is not the same as provoked VTE risk.

-       There is no 3.2 section?

-       Pathophysiology of obesity and related VTE should be described more, in particular the PAI-1 pathway in this specific population.

Reviewer 4 Report

very interesting work. comprehensive review, which addresses many pieces of the puzzle. In my opinion, it should also be emphasized in the discussion that even for distal thrombosis there is no equality of views and results. I would add this remark in the discussion.

Round 2

Reviewer 1 Report

Dear Authors,

Thankyou very much for the responses. All comments by the reviewers have been addressed. I recommend the manuscript to be accepted.